

# Signal-to-noise ratio in diffusion-ordered spectroscopy: how good is good enough?

Jamie Guest[1], Peter Kiraly[1,2], Mathias Nilsson[1], Gareth A. Morris[1]

[1]Department of Chemistry, University of Manchester, Oxford Road, Manchester, M13 9PL, UK
[2]JEOL UK Ltd., Bankside, Long Hanborough, OX29 8SP, UK

*Correspondence to*: Gareth A. Morris (g.a.morris@manchester.ac.uk)

**Abstract.** Diffusion-ordered NMR spectroscopy (DOSY) constructs multidimensional spectra displaying signal strength as a function of Larmor frequency and of diffusion coefficient from experimental measurements using pulsed field gradient spin or stimulated echoes. Peak positions in the diffusion domain are determined by diffusion coefficients estimated by fitting
experimental data to some variant of the Stejskal-Tanner equation, with the peak widths determined by the standard error estimated in the fitting process. The accuracy and reliability of the diffusion domain in DOSY spectra are therefore determined by the uncertainties in the experimental data, and thus in part by the signal-to-noise ratio of the experimental spectra measured. Here the Cramér-Rao lower bound, Monte Carlo methods and experimental data are used to investigate the relationship between signal-to-noise ratio, experimental parameters, and diffusion domain accuracy in 2D DOSY experiments.
Experimental results confirm that sources of error other than noise put an upper limit on the improvement in diffusion domain accuracy obtainable by time averaging.



## 1 Introduction

The utility of pulsed field gradient spin or stimulated echo (PFGSE) experiments for distinguishing between the NMR signals of different species was first pointed out by Stilbs (Stilbs, 1981), but practical applications of this principle only became common with the introduction of diffusion-ordered spectroscopy (DOSY) by Morris and Johnson (Morris, 1992). In DOSY (Johnson, 1999; Morris, 2007), a pseudo-2D (or higher dimensional) spectrum is synthesised in which the signals of an NMR spectrum are dispersed into an extra dimension according to the estimated diffusion coefficient $D$. This is obtained by fitting

experimental measurements of signal attenuation as a function of pulsed field gradient amplitude to a theoretical model, usually some variation on the Stejskal-Tanner equation. (Stejskal, 1965; Sinnaeve, 2012) The value added by the DOSY approach over simple PFGSE measurements is that since all signals from spins within a given species should show the same diffusion, in favourable cases cross-sections through the DOSY spectrum at different $D$ values give separate spectra – which can be interpreted just as normal 1D spectra – for each of the components of a mixture. This paper examines the impact of one crucial

determinant of the success or failure of a DOSY experiment, the signal-to-noise ratio (SNR) of the experimental data.

One common analogy is that DOSY is akin to performing chromatography within an NMR tube, separating spectra rather than physically separating analytes. The name DOSY is, however, misleading in some respects. In conventional 2D NMR methods such as COSY, NOESY and TOCSY the 2D spectrum can be obtained by direct Fourier transformation of signals that are

phase or amplitude modulated as a function of an evolution period $t_1$. The frequency $F_1$ at which a given signal appears is determined directly by the frequency of evolution in $t_1$: while the phase or amplitude of a signal may behave unexpectedly, the frequency is determined directly by the quantum mechanics, so signals should always appear at the "correct" frequency. In pseudo-2D methods such as DOSY (and relaxation-based analogues, often referred to as relaxation-ordered spectroscopy, ROSY (Lupulescu, 2003; Gilard, 2008; Nishiyama, 2010; Dal Poggetto, 2017)) this is not the case: the diffusion dimension is

a statistical construct, and the positions of signals in the diffusion dimension are scattered about the true $D$ values. When a DOSY spectrum is constructed, peaks in the diffusion domain are conventionally given Gaussian shapes with widths that reflect the uncertainty in $D$ estimated from the fitting statistics. Thus in COSY spectra, peaks with the same chemical shift are exactly aligned; in DOSY spectra, peaks with the same diffusion coefficient have Gaussian shapes that should overlap but are not coincident. This is just one reason why the interpretation of DOSY spectra demands more of the spectroscopist's skill and

judgment than most other types of NMR spectrum; others include the effects of signal overlap and of systematic errors introduced by imperfect experiments.

In simple mixtures in which the NMR signals are well resolved and the individual species have very different diffusion coefficients, even a crude DOSY experiment will work well. Where species of similar size, and hence similar $D$, are to be

resolved, however, high quality experimental data are essential. One of the key determinants of the utility of a DOSY spectrum is its diffusion resolution, the minimum difference in $D$ that can safely be distinguished. In the absence of systematic error, this is determined by the signal-to-noise ratio of the experimental data. Here we use theory, empiricism and simulated and experimental data to answer some key questions: how good do our experimental data need to be to resolve a given difference in D? how is the uncertainty in $D$ related to the signal-to-noise ratio (SNR) of raw experimental data, and can this relationship

be expressed in a simple form? and at what point do improvements in SNR stop translating into improved resolution in the diffusion domain?

## 2 Methods

In its commonest ("high resolution") form, DOSY uses least squares fitting of the amplitudes of peaks in pulsed field gradient echo spectra to determine diffusion coefficients $D$. A series of $N$ otherwise identical experiments is carried out in which the

amplitudes $G$ of diffusion-encoding field gradient pulses are varied to map out the decay of signal amplitude as a function of



$G$. In the great majority of experiments, a simple fit to a single exponential is used; multiexponential fitting is possible, but is extremely demanding of SNR (Nilsson, 2006) and is not considered here. The diffusional attenuation $S_i/S_0$ in successive measurements takes the form

$\quad S_i/S_0 = \exp(-b_i D)$ (1)

where the form of $b_i$ is determined by the pulse sequence used (Sinnaeve, 2012). In the simple case of a pulsed field gradient spin or stimulated echo in which spatial encoding and decoding are performed by two gradient pulses of duration $\delta$ a time $\Delta - \delta$ apart,


$\quad b_i = \gamma^2 G_i^2 \delta^2 (\Delta - \delta/3)$ (2)

if the gradient pulses are rectangular in shape, or

$\quad b_i = \gamma^2 G_i^2 \delta^2 (\Delta - \delta/4)$ (3)

if half-sine shaped gradient pulses are used. These expressions assume that the field gradient is constant across the sample, which is not always a good approximation; the effects of field gradient non-uniformity can be taken in to account by replacing the term $G^2$ by an appropriate power series in $G^2$ (Connell, 2009).


Experimental data are imperfect, most notably because of the presence of a background of random electronic noise. In a well-conducted experiment the effect of this on the measurement of the amplitude $S$ of a signal, whether in terms of peak height or of signal integral, is well described by the addition of a Gaussian distribution of standard deviation $\sigma_S$. In the case of peak height, the signal-to-noise ratio (SNR) is by convention defined as $S/(2\,\sigma_S)$ in NMR spectroscopy. In a DOSY dataset using $N$

different gradient strengths $G_i$, each of the $N$ measurements $S_i$ of the amplitude of a given peak will have the same standard deviation $\sigma_S$. The effect of this uncertainty on the value of $D$ determined by nonlinear least squares fitting can easily be found by brute force Monte Carlo simulation, or directly from the Cramér-Rao lower bound (CRLB). The latter has been extensively used in NMR, notably for selecting "optimum" sampling patterns $G_i$ for the simultaneous determination of the diffusion coefficients of species of different $D$ or for the estimation of diffusion distributions $S(D)$ (see e.g. Brihuega-Moreno, 2003;

Franconi, 2018; Reci, 2020; note that the derivations given in the first two references contain some minor typographical errors). Here we use the CRLB for the much more pedestrian purpose of quantifying diffusion resolution in DOSY.

A convenient measure of resolution $R_D$ in the diffusion dimension of the DOSY spectrum is the inverse of the coefficient of variation of $D$, that is the ratio of the estimated $D$ to its estimated standard deviation $\sigma_D$. Using the conventional definition of

SNR given above, expression (10) of (Franconi, 2018) becomes

$$R_D = \left(\frac{D}{\sigma_D}\right) = 2\,SNR\sqrt{\frac{A\,C - B^2}{A}} \quad (4)$$

where


$$A = \int_{i=1}^{N} e^{-2\epsilon_i}\,, \quad B = \int_{i=1}^{N} t_i e^{-2\epsilon_i}\,, \quad C = \int_{i=1}^{N} \epsilon_i^2\, e^{-2\epsilon_i}\,, \text{ and } \epsilon_i = b_i D \quad (5)$$



For a given diffusion coefficient $D$ and choice of $N$ gradient values $G_i$, therefore, the dependence of the resolution $R_D$ on the signal-to-noise ratio of a given signal can be calculated. Here $R_D$ was evaluated as a function of the number $N$ of gradient

values sampled, the maximum exponent $\epsilon_{max}$, and the form of the sampling scheme.

Expressions (4) and (5) allow direct calculation of $R_D$. Equivalent results can be obtained easily by Monte Carlo methods, constructing an attenuation table $e^{-\epsilon_i}$ and then repeatedly adding Gaussian noise $n$ of standard deviation $\sigma_S = 1/(2\,SNR)$ to each point of the table and fitting it to a function of the form $\alpha\,e^{-\beta\epsilon_i}$. The standard deviation $\sigma_\beta$ of the parameter $\beta$ is then the inverse

of $R_D$. Again $R_D$ was evaluated as a function of the number $N$ of gradient values sampled, the maximum exponent $\epsilon_{max}$, and the form of the sampling scheme.

Experimental $^1$H DOSY data were acquired for a 100 mM solution of quinine in DMSO-d$_6$, with 50 mM sodium trimethylsilylpropionate (TSP) as reference, using the Oneshot pulse sequence (Pelta, 2002) on a 500 MHz Varian VNMRS

spectrometer equipped with a 5 mm triaxial gradient probe at 25 °C nominal temperature. 12 quadratically-spaced (equally spaced in gradient squared) nominal gradient amplitudes from 12.5 to 52.8 G cm$^{-1}$ were used, with a net gradient-encoding rectangular pulse width of 1 ms and a diffusion delay $\varDelta$ of 0.16 s. 8 transients of 16384 complex points were acquired for each gradient value in a total experiment time of 5 min. The data were subjected to standard DOSY processing in VnmrJ, consisting of zero-filling, reference deconvolution (Morris, 1997) with a target Lorentzian linewidth of 1.3 Hz, baseline correction, peak

picking, fitting to a Stejskal-Tanner equation modified to compensate for the measured gradient non-uniformity of the probe used (Connell, 2009), and construction of the DOSY spectrum using the fitted signal amplitude, diffusion coefficient $D$, and standard error $\sigma_D$. The signal decay for the quinine methoxy peak at 3.9 ppm, which had a SNR of 14400:1 at the lowest gradient used, was extracted, and the Stejskal-Tanner fit repeated with different additions of synthetic Gaussian noise to investigate the influence of SNR on $R_D$.

**3 Results and Discussion**

Equation (4) shows that, as is intuitively reasonable, the diffusion resolution is directly proportional to $SNR$ (provided that systematic sources of error are negligible). The proportionality constant is, however, a complicated function of the choice of sampling function and its relation to the diffusion coefficient: the more data points are measured the better $R_D$ will be, but just how good depends on what parts of the attenuation curve those points sample. If only the early part of the curve is sampled

($\epsilon_{max} < 1$, where $\epsilon_{max}$ is the maximum value of $\epsilon$) then the effect of diffusion on the measured points will be small, or if too wide a range of gradients $G$ is sampled ($\epsilon_{max} \gg 1$) then many of the measured points will contain very little signal, and in both cases $R_D$ will suffer. In a typical high resolution DOSY experiment, the sample will contain species of different sizes with a range of diffusion coefficients $D$. Where the range is not too wide it is common practice to use a simple sampling scheme in which the field gradient pulse amplitude is incremented either linearly, from some minimum value $G_{min}$ to a maximum $G_{max}$

in equal steps of $G$:

$$G_i = G_{min} + (i-1)(G_{max} - G_{min})/(N-1) \tag{6}$$

or quadratically, from $G_{min}$ to $G_{max}$ in equal steps of $G^2$:


$$G_i = \sqrt{G_{min}^2 + (i-1)(G_{max}^2 - G_{min}^2)/(N-1)} \tag{7}.$$

Because the diffusion-encoding gradient pulses $G$ also play a part in determining coherence transfer pathways in many NMR methods for measuring diffusion, complementing and reinforcing the effects of phase cycling, it is important in practice that



small values of $G_{min}$ be avoided. This is particularly important if experiments such as Oneshot (Pelta, 2002) that employ unbalanced bipolar gradient pulse pairs are used with low numbers of transients (and hence incomplete phase cycling). Common practice is therefore to use a constant ratio $G_{min}/G_{max} = \kappa$, where $\kappa = 0.05 - 0.25$, so that $G$ varies from $\kappa\, G_{max}$ to $G_{max}$. Linear and quadratic sampling give similar diffusion resolution, as is shown below. Quadratic sampling can make it easier to detect systematic deviations from exponential decay as a function of gradient squared, and hence to identify peaks in which

the signals of species of different $D$ overlap.

For a given set of experimental delays and pulse durations, linear and quadratic spacing in $G$ will give different sets of Stejskal-Tanner exponents $\epsilon_i$. Different diffusion coefficients $D$ will give different maxima $\epsilon_{max}$, and because the attenuation caused by the minimum gradient $G_{min}$ depends on $D$, the minimum Stejskal-Tanner exponent $\epsilon_{min}$ will vary slightly with $\epsilon_{max}$. Thus for

linear sampling the Stejskal-Tanner exponents are

$$\epsilon_i = \left[ \kappa + \frac{(i-1)(1-\kappa)}{(N-1)} \right]^2 \epsilon_{max} \tag{8}$$

and for quadratic sampling


$$\epsilon_i = \left[ \kappa^2 + \frac{(i-1)(1-\kappa^2)}{(N-1)} \right] \epsilon_{max} \tag{9}.$$

Figure 1 compares the results of Monte Carlo simulations (small filled circles) of exponential fits for the two sampling schemes, with $SNR = 100$ and $\kappa = 0.05$ in both cases, as a function of $N$ and $\epsilon_{max}$ with the Cramér-Rao upper bounds (open circles) for

$R_D$. As expected, there is excellent agreement between the Monte Carlo and analytical results. The lines for linear regression confirm that there is a direct proportionality with $\sqrt{(N-1)}$ for low $\epsilon_{max}$, but that for higher $\epsilon_{max}$, where the signal is strongly attenuated for greater $\epsilon_i$ values, the line of best fit is displaced. The slope of the line of best fit for $R_D$ as a function of $\sqrt{(N-1)}$ rises as $\epsilon_{max}$ increases until it reaches a maximum at around $\epsilon_{max} = 2.1$, after which it decreases again. This is again as expected: for low $\epsilon_{max}$ the data are dominated by points that have high precision but low attenuation, while for high $\epsilon_{max}$ the converse is

true.

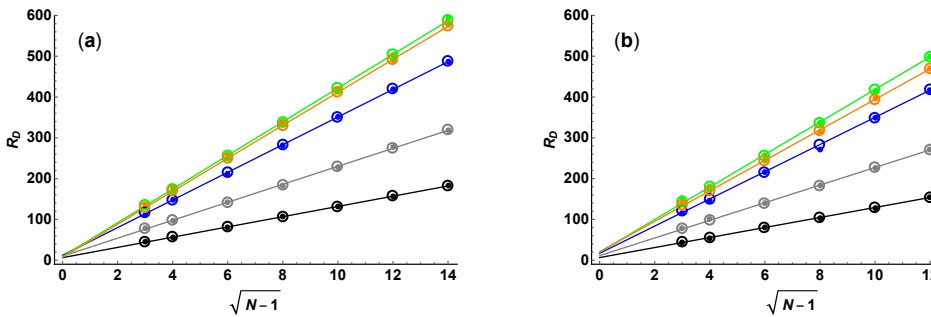

**Figure 1:** Diffusion resolution $R_D$ as a function of $\sqrt{(N-1)}$, where $N$ is the number of gradient values used, for (a) linear and (b)
quadratic sampling in the gradient domain, determined by Monte Carlo simulation (small filled circles) and Cramér-Rao Least Bounds analysis (open circles), for $SNR = 100$ and maximum Stejskal-Tanner exponents $\epsilon_{max}$ of 0.25 (black), 0.5 (grey), 1 (blue), 2 (green), and 3 (orange). Solid lines show the results of linear regression of the Cramér-Rao data.





The predicted diffusion resolution $R_D$ is a function of the sampling scheme, signal-to-noise ratio $SNR$, maximum Stejskal-Tanner exponent $\epsilon_{max}$, and number of gradient values used $N$. Given the nature of Eqs. (4) and (5) it is clear that no simple analytical form exists for $R_D(SNR, \epsilon_{max}, N)$. Equally, it is known that $R_D$ is directly proportional to $SNR$, and it is reasonable to expect $R_D$ to be proportional to the square root of $N - 1$, since (a) increasing $N$ will decrease the effects of random errors in proportion to the square root of the effective number of independent measures of $D$, and (b) that number will be dependent on $N - 1$, since it is the *change* in signal amplitude that provides information on $D$, reducing the number of degrees of freedom by one. In general the effective number will be less than $N - 1$ for all but low values of $\epsilon_{max}$, because signal attenuation will reduce the information content for higher values $\epsilon_i$. It is thus reasonable to seek an approximate analytical representation of the form

$$R_D(SNR, \epsilon_{max}, N) \approx SNR \sqrt{(N - 1)} f(\epsilon_{max}) \tag{10}.$$

Figure 2 shows the variation of $f$ as a function of $\epsilon_{max}$, calculated numerically using Eqs. (4), (5), (8) and (9) for values of $N$ between 10 and 200 for linear and quadratic sampling, together with fits to a three-parameter function of the form

$$f(\epsilon_{max}) = a\,\epsilon_{max}e^{-b\,(\epsilon_{max})^c} \tag{11}.$$

The quality of fit is more than adequate for practical use, establishing a simple relationship between diffusion resolution, signal-to-noise ratio and experimental parameters; fit parameters are given in Table 1.

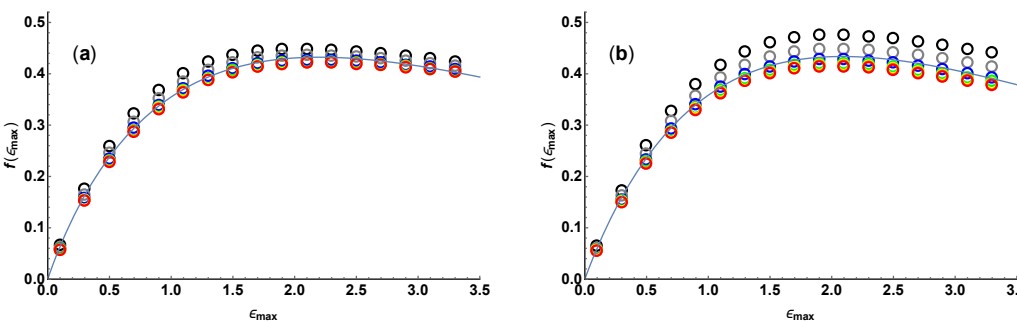

Figure 2: Relative diffusion resolution $f(\epsilon_{max})$ determined Cramér-Rao Least Bounds analysis (open circles) as a function of maximum Stejskal-Tanner exponent $\epsilon_{max}$, for (a) linear and (b) quadratic sampling in the gradient domain with 10 (black), 17 (grey), 37 (blue), 65 (green), 101 (yellow), and 197 (red) gradient values. Solid lines show the results of nonlinear regression of the data points shown to the three-parameter function Eq. (11).

| | Linear sampling | Quadratic sampling |
|---|---|---|
| $a$ | 0.72 | 0.66 |
| $b$ | 0.71 | 0.61 |
| $c$ | 0.77 | 0.86 |

Table 1. Fitted parameters for Eq. (11) obtained from the data of Fig. 2. No error estimates are given as the data fitted are not normally distributed.



In principle, diffusion accuracy should increase indefinitely as the signal-to-noise ratio of the experimental data increases. In practice this is not the case, because spectral noise is far from the only source of uncertainty in the signal attenuations measured in DOSY experiments. Radiofrequency pulse irreproducibility, field-frequency ratio instability, gradient noise, temperature variation and a range of other sources all limit the reliability of signal intensity measurements in NMR, limiting resolution in DOSY and causing $t_1$-noise in multidimensional spectra (Mehlkopf, 1984; Morris, 1992). In general, the accuracy and

reproducibility of NMR data tend to deteriorate as the number of pulses used in a sequence increases (because of pulse phase and amplitude jitter caused by limited radiofrequency spectral purity), as the durations of the delays used increase (because of the cumulative effect of field-frequency fluctuations), and as the overall duration of an experiment increases (because of slow changes in environmental factors such as room temperature, air pressure etc.). Most such perturbations are at least semi-systematic in nature, but many (particularly pulse phase instability) have effects that can appear random, and can therefore

decrease, at least to some extent, with time averaging. Other sources of distortion in the measured signal decay are both systematic and reproducible and therefore do not decrease with time averaging. These include changes in signal attenuation caused by convection (never wholly absent in practical NMR experiments on liquids (Swan, 2015; Barbosa, 2016)), and by the presence of signals from unwanted coherence transfer pathways. Distortions caused by spatial non-uniformity of the field gradient can be corrected for if appropriate calibration is performed (Connell, 2009).


There is thus a practical limit to the benefits to be gained by increasing SNR, whether by time averaging, increasing the signal strength (e.g. by increasing sample concentration), or reducing the noise (e.g. by using a cold probe and preamp). This is illustrated here with experimental data obtained as described earlier for the methoxy signal from a sample of quinine. The starting SNR of the quinine methoxy peak in the lowest gradient spectrum was 14400:1; successively greater amounts of

synthetic Gaussian noise were added and fitting repeated, averaging the results of 100 additions, to show the influence of SNR on the diffusion resolution $R_D$. If the contributions from sources other than noise to the errors in the experimental peak height as a function of gradient strength are normally distributed and have a root mean square deviation which is a fraction $1/(2\,SNR_{lim})$ of the initial peak amplitude, then the effect on fitting, and hence on diffusion resolution, of adding uncorrelated noise is to degrade the effective signal-to-noise ratio $SNR$ in Eq. (10) to

$$SNR_{eff} = SNR \sqrt{\frac{SNR_{lim}^2}{SNR_{lim}^2 + SNR^2}} \quad\quad\quad\quad\quad (12).$$

This gives a final predicted diffusion resolution for given experimental conditions of

$$R_D(SNR, \epsilon_{max}, N) \approx \frac{SNR}{\sqrt{1 + \left(\frac{SNR}{SNR_{lim}}\right)^2}} \sqrt{(N-1)}\, f(\epsilon_{max}) \quad\quad\quad\quad (13),$$


where $f(\epsilon_{max})$ can be approximated by Eq. (11). Thus if the noise contribution to the experimental uncertainty is dominant, the effective signal-to-noise ratio is the actual SNR, but at high SNR the effective signal-to-noise ratio for the purposes of Stejskal-Tanner fitting is the limit $SNR_{lim}$ imposed by other error sources.





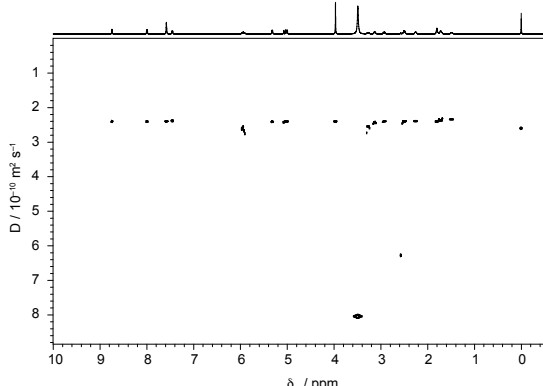

**Figure 3: 500 MHz Oneshot $^1$H DOSY spectrum of 100 mM quinine in DMSO-d$_6$ with 50 mM sodium trimethylsilylpropionate as reference, acquired as described in the text.**

To investigate the effect of signal-to-noise ratio on diffusion resolution, synthetic noise was added to the experimental data used to construct the $^1$H DOSY spectrum of quinine shown in Fig. 3. Figure 4 shows the effect of SNR on the measured $R_D$

for experimental data for the quinine methoxy peak, found by titrating in extra noise as described above. The experimental signal-to-noise ratio of the first gradient increment was 14400:1, but the diffusion resolution $R_D$ found when the raw experimental data were fitted was only 420, a small fraction of the predicted value of almost 15000. As Figure 3 shows, at low SNR the observed diffusion resolution follows the line expected for the unmodified Cramér-Rao limit of Eq. (11), but as SNR increases the improvement in $R_D$ levels off, slowly approaching the limit seen for the data with no noise added. Fitting of Eq.

(13) to the noise-supplemented experimental data gave a value of just over 300 for $SNR_{lim}$. To put this value in context, it corresponds to a respectably small root mean square uncertainty in the signal amplitudes measured of $1/600 \sim 0.17\%$ of the original peak intensity, typical of good quality results obtained with multiple pulse sequences on a modern spectrometer. With extended time averaging and appropriate precautions and instrumental interventions it is possible to obtain data with significantly smaller uncertainties than this (see e.g. Power, 2016), but the cost in time and effort can be considerable.


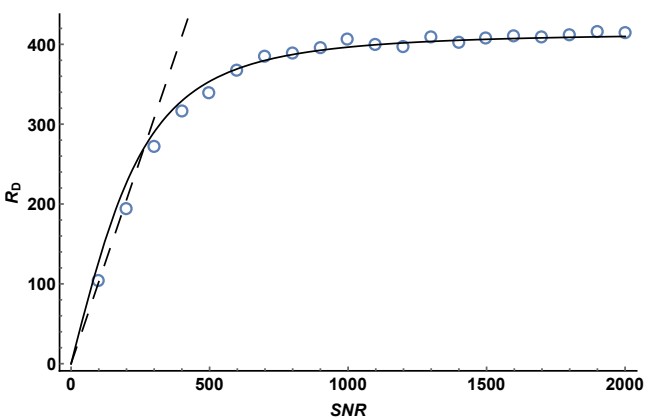

**Figure 4: Diffusion resolution $R_D$ as a function of signal-to-noise ratio for the methoxy signal of quinine in a Oneshot experiment on a 100 mM solution of quinine in DMSO-d$_6$. Open circles show the average of 100 values of $R_D$ by fitting of the experimental data with the addition of synthetic Gaussian noise for each value of SNR, the dashed line shows the predicted Cramér-Rao limit, Eq. (11),**

**for the experimental parameters used ($N = 12$, $\epsilon_{max} = 0.76$), and the solid line the result of nonlinear least-squares fitting of the Cramér-Rao limit modified to take into account the presence of other errors in the signal intensity, Eq. (12), with $SNR_{lim} = 305$.**



## 4 Conclusions

It is well known that the signal-to-noise ratio of diffusion-weighted experimental NMR data plays a critical role in determining the diffusion resolution of a DOSY spectrum constructed from it. There is thus a temptation to conduct very long experiments with extensive time averaging in order to obtain the best possible results. Conversely, in dilute systems the temptation is to conduct equally long experiments in the hope of obtaining results with sufficient diffusion resolution to shed light on speciation etc. In both cases it is possible, and indeed common, to waste a great deal of instrument time for no good result, either because sources of error other than noise dominate the fitting statistics, or because the final signal-to-noise ratio is insufficient. Here it

is shown that a trivial calculation will show both whether or not such experiments are worth attempting in the first place, and what limiting diffusion resolution is achievable.

### Code and data availability

Raw experimental data for Fig. 3 and the Mathematica code used to generate Figs. 1, 2 and 4 can be downloaded from DOI
10.17632/d7bdxz9hsk.1.

### Author contributions

GAM and MN designed the experiments and simulations. JG and PK carried out the experimental work. JG and GAM performed the simulations and analysis. GAM prepared the manuscript with contributions from all co-authors.


### Competing interests

Mathias Nilsson is an editor of MR.

### Acknowledgments

This work was supported by the Engineering and Physical Sciences Research Council (grant numbers EP/N033949/1 and EP/R018790).

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
