# Peer review of "Signal-to-noise ratio in diffusion-ordered spectroscopy: how good is good enough?"

_Magnetic Resonance, 2021_

## Author Response (AR1)

Referee 1

We are grateful to both referees for their helpful comments and for noting several errors.

An important problem in DOSY is how well differences in diffusion coefficient can be resolved. In older literature, a "rule of thumb" can be found stating that (in case of nonoverlapping signals and good signal-to-noise ratios) the diffusion coefficients have to differ by at least 30% to be distinguished, but the reality is of course more complex.

*This statement is a little confused. In the limit of good SNR and no overlap, differences in D as small as 1% can be distinguished in practice. The 30% figure is commonly quoted as the resolution limit for biexponential fitting where signals are overlapped, although at very high SNR signals of comparable intensity can be resolved with somewhat smaller differences (Anal Chem 78, 3040).*

This work presents a very welcome quantitative assessment of how the signal to noise and sampling of gradient strengths affect the diffusion resolution. These new insights can indeed help practitioners to assess beforehand whether it will at all be feasible to resolve different molecules in the diffusion dimension, using for instance also tools to predict diffusion coefficients (based on molecular weight, by the same research group).

I do have some questions that the authors may wish to clarify or consider commenting on.

The final equation (13) illustrates that in practice improving the diffusion resolution by increasing the signal-to-noise ratio has its limits. The same equation seems to suggest that an increase in the number of gradient values N, rather than increasing the number of transients, could indefinitely improve the diffusion resolution. Figure 2 indeed shows no deviation from the linear behaviour of $R_D$ as a function of $\sqrt{(N-1)}$. Do the authors think that in reality there is also here a limit to be reached, for instance due to gradient hardware limitations, or environmental changes as a function of time or gradient strength?

*There are two points at issue here. First, the analysis in this manuscript specifically excludes the influence of such systematic errors, which will indeed impose limits on diffusion resolution. Second, Fig. 2 only describes the effect of spectral noise on diffusion resolution. As is explained later in the manuscript, random and pseudorandom perturbations of the measured signal intensity from other sources, for example pulse irreproducibility, do indeed also limit diffusion resolution. The effect of these random and pseudorandom perturbations averages out with increasing N just as the effect of noise averages out with increasing time averaging. What is left in the limit of infinite numbers of scans and of gradient values is the effect of systematic error. In this unattainable limit, and with no complications such as peak overlap, conventional DOSY processing would lead to diffusion peaks with finite widths, determined by the systematic deviation of the measured data from the Stejskal-Tanner equation used, but always at the same apparent D.*

The value of $SNR_{lim}$ in equation (13) appears to be determined by systematic errors in signal intensity, which, besides hardware and environment fluctuations, will probably depend on the pulse sequence used. The authors rightly mention that in general more rfpulses in the sequence or additional unwanted coherence transfer pathways will result in more systematic 'noise'. I wonder if $SNR_{lim}$, which can be determined experimentally in the way described in the paper, could serve as a means to compare the performance of various DOSY pulse sequences, comparing it to, for instance, the value measured for the oneshot sequence on the same spectrometer and sample?

*It is important to distinguish here between systematic and experimentally reproducible sources of error (e.g. gradient non-uniformity) and irreproducible random or pseudorandom sources of error (e.g. gradient noise, pulse irreproducibility). This manuscript deals with the latter: the former is a different can of worms, and has been addressed elsewhere (including the references by Connell et al and Damberg et al.). The limit imposed by $SNR_{lim}$ derives from random/pseudorandom variations in signal intensity; it could be used in comparing pulse sequences, but the choice of what sequence to use in a given context also depends on a range of other factors.*

Figure 2 shows that the data points obtained for low values of N (10 (black) and 17 (grey)) deviate somewhat more from the fitted curve than all the other data points. Does this imply that equation (13), combined with equation (11) and Table 1, approximates reality less well for lower values of N?

*That is correct.*

Some further technical comments that should be fixed:
In equation (3), the gradient shape factor for half-sine shapes, $(2/\pi)^2$, has been forgotten.

*To be clarified on revision. [Bruker's Topspin software, which is used for almost all acquisitions using half-sine pulses, defines an effective gradient $G_i = G_{max}(2/\pi)$]*

Equation (5), expression for B, shows $t_i$ before the exponent. I guess this should be $\varepsilon_i$ .?

*To be corrected on revision.*

There are problems with the references. Some citations in the main text do not feature in the reference list (I spotted Brihuega-Moreno 2003, Franconi 2018, Reci 2020 and Power 2016 to be lacking). The reference to Mehlkopf et al. lacks the title.

*To be corrected on revision.*

Referee 2

We are grateful to both referees for their helpful comments and for noting several errors.

In this article, Guest et al. analyse the effect of the signal to noise ratio in diffusion-ordered NMR spectroscopy (DOSY), and provide guidelines on a choice of sampling strategy (number of gradient increment, maximum attenuation, SNR) that provides good accuracy. The paper is very well written and contains a number of interesting and useful explanations on DOSY. The main message is enlightening and important for users of the method. I recommend publication in Magnetic Resonance, after the following minor points have been addressed.

It would be useful to clarify what is meant by "accuracy" in the text. Sometimes the word refers to systematic errors only, sometimes to a combination of systematic and random errors (https://en.wikipedia.org/wiki/Accuracy_and_precision). Here the latter seems to be used, but this would need to be explicit.

*This is not straightforward. The diffusion dimension of a DOSY spectrum plots an estimated probability distribution of values of D: each resonance has a Gaussian peak centred on the estimated diffusion coefficient with a width determined by the estimated standard error obtained in the fitting process. Viewed simply, the width of the diffusion peak is thus a measure of its experimental precision, and the difference between the peak position and the true D is a measure of its accuracy - just as is illustrated in the figure on the Wikipedia page. On a strict view, however, the difference between the peak position and the true D in the absence of systematic error is purely a reflection of the precision of the estimate of D obtained by fitting: if a large number of DOSY experiments were performed, all perfect apart from the effect of noise, the accuracy would be infinite (the average of the estimated D values would converge on the correct value). The ambiguity arises because of the nature of the DOSY display. We took the pragmatic view that while it is a loose usage, accuracy, in the ISO sense of trueness, is the more helpful word for readers. To be clarified on revision.*

The conclusion reads "a trivial calculation will show both whether or not such experiments are worth attempting in the first place, and what limiting diffusion resolution is achievable". Does this calculation require knowledge of SNR_lim? How can this quantity be

determined?

*SNR_lim is not needed to determine, using Eq. (11), whether it is possible for a given experiment under otherwise ideal conditions to achieve sufficient SNR in the time available to give the diffusion resolution required. A more sophisticated calculation, using Eq. (13), would show whether instrumental limitations would impose a more restrictive limit, but this would require experimental characterisation of the effects of instrumental irreproducibility.*

*To be clarified on revision.*

It would be useful to have guidelines on what to do in a fixed total experimental time. Is it better to increase the number of gradient increments, or the number of averaged scans? In which cases? The answer lies in the proposed equations, but this is so frequent a question that it may deserve a specific discussion. For example, it seems from Eq. 13, that increasing N will always increase accuracy, while increasing SNR is only useful up to a certain limit. Is it the case that one should increase N only as soon as the number of scans is sufficient for phase cycling purposes and peak detectability ?

*The short answer is that in the case analysed in this manuscript (no systematic errors), increasing SNR will only be useful up to a limit set by N, and increasing N will only be useful up to a limit set by SNR, but that there is no limit to the accuracy obtainable by increasing both N and SNR. The long answer is that of course systematic errors play a crucial role in limiting DOSY, and that in choosing experimental parameters there are many other factors to be taken into account (expected range of D, the desirability of being able to detect multiexponential decay, the variation of signal irreproducibility with signal amplitude, the effect of $B_1$ inhomogeneity, the effects of electrical nonlinearity of the gradient circuitry and spatial nonlinearity of the gradients applied, ...). The question of optimum sampling strategy is beyond the scope of this manuscript, which deals purely with the effect of SNR. (See also the reply to the first point raised by Referee 1).*

Overall, while all the tools are provided to guide readers in the choice of appropriate parameters, the usefulness of the paper would be increased by the addition of a practical example.

*Figures 3 and 4 provide this.*

The "inverse of the coefficient of variation" is introduced as "a convenient measure of resolution". This choice should be justified. In spectroscopy, resolution or dispersion is usually quoted on an absolute, not a relative scale. Why use a relative scale here ?

*This is straightforward. In the diffusion dimension, unlike the spectral dimension, linewidths scale with D. A 1% error in a D of 1 x 10–10 m2/s has a tenth the impact of a 1% error in a D of 10 x 10–10 m2/s. This is (partly) why when DOSY spectra are conventionally plotted, with D increasing from top to bottom, the linewidths in the diffusion dimension also tend to increase from top to bottom.*

In Eq. 4, the half sine shape seems to be accounted for in Deltaprime, but not in the gradient area.

*To be clarified on revision. [Bruker's Topspin software, which is used for almost all acquisitions using half-sine pulses, defines an effective gradient $G_i = G_{max}(2/\pi)$]*

In Eq. 5, shouldn't a sum symbol be used instead of an integral symbol?

*It certainly should! To be corrected on revision..*

Also in Eq. 5: what is the variable t_i ? From Franconi et al., it should be espilon_i ?

*To be corrected on revision.*

I could not find the reference to Reci et al. and Franconi et al. in the manuscript

*To be corrected on revision.*